# Ixazomib, Lenalidomide, and Dexamethasone (IRD) Treatment with Cytogenetic Risk-Based Maintenance in Transplant-Eligible Myeloma: A Phase 2 Multicenter Study by the Nordic Myeloma Study Group

**DOI:** 10.3390/cancers16051024

**Published:** 2024-02-29

**Authors:** Anu Partanen, Anders Waage, Valdas Peceliunas, Fredrik Schjesvold, Pekka Anttila, Marjaana Säily, Katarina Uttervall, Mervi Putkonen, Kristina Carlson, Einar Haukas, Marja Sankelo, Damian Szatkowski, Markus Hansson, Anu Marttila, Ronald Svensson, Per Axelsson, Birgitta Lauri, Maija Mikkola, Conny Karlsson, Johanna Abelsson, Erik Ahlstrand, Anu Sikiö, Monika Klimkowska, Reda Matuzeviciene, Mona Hoysaeter Fenstad, Sorella Ilveskero, Tarja-Terttu Pelliniemi, Hareth Nahi, Raija Silvennoinen

**Affiliations:** 1Department of Medicine, Kuopio University Hospital, 70210 Kuopio, Finland; 2Department of Hematology, St. Olavs Hospital, 7030 Trondheim, Norway; anders.waage@ntnu.no; 3Hematology, Oncology and Transfusion Medicine Center, Vilnius University Hospital, 08661 Vilnius, Lithuania; valdas.peceliunas@santa.lt; 4Oslo Myeloma Center, Department of Hematology, Oslo University Hospital, 0450 Oslo, Norway; fredrikschjesvold@gmail.com; 5KG Jebsen Center for B Cell Malignancies, University of Oslo, 0316 Oslo, Norway; 6Helsinki University Hospital Cancer Center Hematology, University of Helsinki, 00029 Helsinki, Finland; pekka.anttila@hus.fi (P.A.); raija.silvennoinen@helsinki.fi (R.S.); 7Hematology-Oncology Unit, Oulu University Hospital Hematology, 90220 Oulu, Finland; marjaana.saily@pohde.fi; 8Medical Unit Hematology, Karolinska University Hospital, 171 64 Solna, Sweden; katarina.uttervall@regionstockholm.se; 9Department of Medicine, Karolinska Institutet, 171 77 Stockholm, Sweden; 10Department of Medicine, Turku University Hospital, 20521 Turku, Finland; mervi.putkonen@tyks.fi; 11Department of Hematology, Uppsala University Hospital, 751 85 Uppsala, Sweden; kristina.carlson@akademiska.se; 12Stavanger University Hospital, 4011 Stavanger, Norway; haer@sus.no; 13Hematology Unit, Department of Internal Medicine, Tampere University Hospital Hematology, 33520 Tampere, Finland; marja.sankelo@pirha.fi; 14Department of Oncology, Hematology and Palliative Care, Foerde Central Hospital, 6812 Foerde, Norway; damian.lukasz.szatkowski@helse-forde.no; 15Department of Hematology, Skåne University Hospital, 222 42 Lund, Sweden; markus.hansson@skane.se; 16Department of Medicine, Kymenlaakso Central Hospital, 48210 Kotka, Finland; anu.marttila@kymenhva.fi; 17Department of Hematology, Linköping University Hospital, 581 85 Linköping, Sweden; ronald.svensson@regionostergotland.se; 18Department of Haematology, Helsingborg Hospital, 252 23 Helsingborg, Sweden; per.axelsson@skane.se; 19Department of Hematology, Sunderby Hospital, 971 80 Luleå, Sweden; birgitta.lauri@norrbotten.se; 20Department of Medicine, Päijät-Häme Central Hospital, 15850 Lahti, Finland; maija.mikkola@paijatha.fi; 21Department of Haematology, Halland Hospital, 302 33 Halmstad, Sweden; 22Department of Hematology, Uddevalla Hospital, 451 53 Uddevalla, Sweden; johanna.abelsson@vgregion.se; 23Department of Medicine, Örebro University Hospital, 701 85 Örebro, Sweden; erik.ahlstrand@regionorebrolan.se; 24Department of Medicine, Central Finland Central Hospital, 40620 Jyväskylä, Finland; anu.sikio@hyvaks.fi; 25Department of Clinical Pathology and Cytology, Karolinska University Hospital, 141 86 Stockholm, Sweden; monika.klimkowska@sll.se; 26Department of Physiology, Biochemistry, Microbiology and Laboratory Medicine, Biomedical Sciences Institute, Vilnius University Hospital and Vilnius University Faculty of Medicine, 03101 Vilnius, Lithuania; reda.matuzeviciene@santa.lt; 27Department of Immunology and Transfusion Medicine, St. Olavs Hospital, 7030 Trondheim, Norway; mona.hoyseter.fenstad@stolav.no; 28Clinical Chemistry, Helsinki University Hospital, University of Helsinki, 00014 Helsinki, Finland; sorella.ilveskero@hus.fi; 29Fimlab Laboratories Ltd., 33520 Tampere, Finland; tarja-terttu.pelliniemi@fimlab.fi; 30Hematology Centre, Karolinska University Hospital Huddinge, 141 57 Stockholm, Sweden; hareth.nahi@ki.se

**Keywords:** multiple myeloma, ixazomib, autologous stem cell transplantation, measurable residual disease, progression-free survival, maintenance

## Abstract

**Simple Summary:**

Outcomes for high-risk myeloma patients are still poor, despite many advances in treatment. In addition, scarce data exist on double maintenance in transplant-eligible high-risk newly diagnosed multiple myeloma (NDMM) patients. We present the results of a prospective study on 120 transplant-eligible NDMM patients with prolonged cytogenetic risk-based all-oral maintenance with lenalidomide + ixazomib (IR) for high-risk patients and lenalidomide (R) alone for non-high-risk patients after ixazomib–lenalidomide–dexamethasone (IRD) induction plus autologous stem cell transplantation followed by IRD consolidation. We found that high-risk cytogenetics had no impact on the proportion of patients achieving sustained undetectable minimal residual disease or on the rate of progression-free survival with IR maintenance. Our data suggest that prolonged use of all-oral double maintenance with IR with reasonable adverse effects would be a potential option for high-risk myeloma patients.

**Abstract:**

Scarce data exist on double maintenance in transplant-eligible high-risk (HR) newly diagnosed multiple myeloma (NDMM) patients. This prospective phase 2 study enrolled 120 transplant-eligible NDMM patients. The treatment consisted of four cycles of ixazomib–lenalidomide–dexamethasone (IRD) induction plus autologous stem cell transplantation followed by IRD consolidation and cytogenetic risk-based maintenance therapy with lenalidomide + ixazomib (IR) for HR patients and lenalidomide (R) alone for NHR patients. The main endpoint of the study was undetectable minimal residual disease (MRD) with sensitivity of <10^−5^ by flow cytometry at any time, and other endpoints were progression-free survival (PFS) and overall survival (OS). We present the preplanned analysis after the last patient has been two years on maintenance. At any time during protocol treatment, 28% (34/120) had MRD < 10^−5^ at least once. At two years on maintenance, 66% of the patients in the HR group and 76% in the NHR group were progression-free (*p* = 0.395) and 36% (43/120) were CR or better, of which 42% (18/43) had undetectable flow MRD <10^−5^. Altogether 95% of the patients with sustained MRD <10^−5^, 82% of the patients who turned MRD-positive, and 61% of those with positive MRD had no disease progression at two years on maintenance (*p* < 0.001). To conclude, prolonged maintenance with all-oral ixazomib plus lenalidomide might improve PFS in HR patients.

## 1. Introduction

Ixazomib is the first oral selective and reversible proteasome inhibitor. It disturbs the ubiquitin proteasome system by inhibiting chymotrypsin-like activity, which results in apoptosis of myeloma cells [1]. Ixazomib is approved as a part of an all-oral three-drug regimen (ixazomib–lenalidomide–dexamethasone) for patients with relapsed or refractory myeloma [2], albeit not in first-line therapy. A phase 2 IFM2013-06 study with 42 patients concluded that induction with IRD followed by autologous stem cell transplantation (ASCT) and IRD consolidation with fixed one-year ixazomib maintenance is safe and effective in NDMM patients with 92.8% 3-year OS [3]. However, the Spanish Pethema group showed that adding ixazomib as part of a maintenance triplet after ASCT did not prolong PFS, possibly due to reduced ixazomib doses caused by toxicity [4]. A placebo-controlled study by Dimopoulos et al. concluded that ixazomib maintenance alone improved PFS compared to placebo [5]. In addition, the results of the MMRC-066 study showed IRD consolidation to be safe and ixazomib to be noninferior to lenalidomide as maintenance after ASCT [6]. Scarce data exist on the combination of ixazomib-based induction, consolidation and maintenance with or without lenalidomide and on MRD responses by flow cytometry or molecular assessment. Of note, Paiva et al. [7] pooled Tourmaline-MM3 and Tourmaline-MM4 data together and found ixazomib to be superior for improvement in PFS compared to placebo in patients with MRD positivity before maintenance.

Measurable residual disease (MRD) assessment by multiparameter flow cytometry (MFC), polymerase chain reaction (PCR) with appropriate ASO probe, targeted mass spectrometry (MS-MRD) or next-generation sequencing (NGS) has been used to evaluate the depth of treatment response in MM patients [8,9,10,11,12]. Patients with undetectable MRD and especially patients with sustained undetectable MRD have proved to be at reduced risk of relapse and death [9,13,14,15]. In the era of novel agents, MRD assessment is recommended at least in clinical studies, although the method of choice might differ between the centers [16,17].

Lenalidomide maintenance is the standard of care after ASCT to prolong PFS and even OS based on the findings of a previous meta-analysis [18]. The phase 3 multicenter Myeloma XI study concluded that the benefit of lenalidomide maintenance also covered high-risk (HR) MM patients with del (17 p,) t(4;14), t(14;16) or 1q gain irrespective of the induction therapy used [13]. However, the optimal duration of lenalidomide maintenance is not known.

The outcome of high-risk patients is still poor, despite many advances in myeloma treatment. We designed this phase 2 prospective Nordic Myeloma Study Group (NMSG) trial to investigate the role of all oral double maintenance with ixazomib + lenalidomide in cytogenetic high-risk (HR) patients. The main endpoint of the study was undetectable minimal residual disease (MRD) with sensitivity of < 10^−5^ by flow cytometry at any time during treatment. The focus was also to evaluate whether the HR patients could reach similar outcomes as non-high-risk (NHR) patients when receiving ixazomib plus lenalidomide as maintenance instead of lenalidomide alone in NHR patients. Here, we present the planned analysis of this study: when the last patient included has been two years on maintenance.

## 2. Patients and Methods

### 2.1. Patients and Treatment

Altogether, 120 NDMM patients intending to proceed to ASCT in 22 NMSG sites in Finland, Norway, Sweden, and Lithuania were included in this prospective, multicenter study. The criteria for inclusion were age between 18 and 70 years, symptomatic and measurable disease diagnosed by International Myeloma Working Group, CRAB or biomarker criteria, and Eastern Cooperative Oncology Group (ECOG) performance status ≤ 2. Included patients were required to have absolute neutrophil count (ANC) ≥ 1000/mm^3^ (≥1.0 × 10^9^/L), platelet count ≥ 75,000/mm^3^ (75 × 10^9^/L), total bilirubin ≤ 1.5 × the upper limit of the normal range (ULN), alanine aminotransferase (ALT) and aspartate aminotransferase (AST) ≤ 3 × ULN and calculated creatinine clearance ≥ 30 mL/min (Cockcroft–Gault estimation of creatinine clearance).

The cytogenetic risk profile that was the basis for the categorization of maintenance treatment was determined before the start of induction treatment. Patients with t(4;14), t(14;16), t(14;20), del (17 p) or 1q gain were included in the HR group [19]. Of note, PFS and OS analyses were also conducted comparing the NHR group to the HR group without 1q gain. All patients received the same evaluations and treatment until maintenance treatment, which was given according to the defined HR/NHR criteria.

The patients received four cycles of ixazomib combined with lenalidomide and dexamethasone (IRD; ixazomib 4 mg on days 1, 8, 15, lenalidomide 25 mg on days 1–21, dexamethasone 40 mg weekly in 28-day cycles) as an induction therapy. The patients with less than partial remission (PR) after induction were excluded. Mobilization of CD34^+^ cells and ASCT were performed according to the local standard practices. Three months after ASCT, all patients received two consolidation cycles with the same regimens as during induction, followed by maintenance consisting of ixazomib 4 mg on days 1, 8, 15 and lenalidomide 10 mg on days 1–21 for HR patients and lenalidomide alone 10 mg on days 1–21 for NHR patients. We intended to increase the dose of lenalidomide to 15 mg in both groups from the fourth cycle onwards. Treatment was continued until progression (PD) or intolerable toxicity.

### 2.2. Methods

Serological responses were assessed before each cycle. Protein electrophoresis, immunofixation and MRD analyses were centralized to the university hospitals. If CR or better were achieved, bone marrow (BM) aspirate was taken for fresh flow cytometric MRD analysis. Sampling was repeated every 6 months. Low-dose body CT was performed three months after ASCT to confirm response.

Fluorescence in situ hybridization (FISH) by standard methods [20] was performed at laboratories of genetics in university hospitals in Finland, Vilnius, Lund, Stockholm and Trondheim. Analyses of isolated CD138^+^ plasma cells were utilized to determine the HR cytogenetic group, which included t(4;14), t(14;16), t(14;20), del (17 p) or 1q gain [19]. For inclusion in the HR group, the cutoffs by FISH were 60% for del (17 p) and 5% for other high-risk cytogenetic findings. CD138 selection was performed before the FISH analysis.

Flow cytometry samples were assayed within 24 h after collection according to the next-generation flow cytometry (NGF) approach [16,21]. The following eight-color panel was employed in the analysis: CD38-FITC (multiepitope), CD56-PE (clone C5.9) and CD81 APCC750 (clone M38) from Cytognos, Salamanca, Spain, CD117-APC (clone 104D2) and CD138-BV421 (clone MI15) from BD Biosciences, San Jose, CA, USA, CD45 PerCP Cy5.5 (clone HI30) from BioLegend (San Diego, CA, USA), CD19 PC7 (clone J3-119) from Beckman-Coulter (Miami, FL, USA) and CD27-BV510 (clone O323) from BioLegend or corresponding MM-MRD kit (Cytognos, Salamanca, Spain) with CD27 and CD138 as drop-in reagents (clones as above with the exception of CD19-PECy7 clone 19-1). Staining of intracytoplasmic kappa and lambda light chains with polyclonal reagents after permeabilization with Fix&Perm (Nordic MUBio, Susteren, The Netherlands) was optional. Either BD FACSCanto II (BD Biosciences, San Jose, CA, USA) or Navios (Beckman-Coulter, Miami, FL, USA) flow cytometers were used. Performance was monitored by electronic quality control surveys. MRD was calculated as percentage of total nucleated cells (TNC). A population of 20 events was the lower limit of detection (LOD) for aberrant plasma cells. The sample was representative if mast cells (CD117^hi^), erythroblasts (CD45^lo^/side-scatter^lo^) and B-cell precursors (CD19+/CD38^hi^/CD45^dim^) were present and at least 2 million nucleated cells per tube were acquired to reach a sensitivity of at least 10^−5^. MRD analyses were undertaken using flow cytometry. CR was confirmed with low-dose CT or MRI three months after ASCT. Monitoring of the study was performed according to level 2: during the initial meeting, after including the first study patient, after the first cycle of the first study patient on-site, and thereafter annually. All study subjects were monitored for adverse and serious adverse events, treatment response, achievement of primary and secondary endpoints, date of progression and withdrawal or discontinuation.

### 2.3. Endpoints

The main endpoint was to determine the proportion of patients with undetectable flow MRD with sensitivity of <10^−5^ at any time during treatment, and other endpoints were overall response rate (ORR), safety, PFS and OS in transplant-eligible NDMM patients.

### 2.4. Statistical Analysis

Efficacy analyses are based on the intention-to-treat population (*n* = 120). Fraction of adverse events is based on the number of patients that started treatment (*n* = 120). The data cutoff was 13 December 2022: when the last patient included had been observed until two years on maintenance. According to A’Hern single-stage design analysis, the power of the study with a total sample size of 120 patients is 80% with 2-sided significance level of alpha 0.05 in order to verify the hypothesis that the new protocol treatment could increase the fraction of patients with MRD level < 10^−5^ from 21%, which was the fraction found in the historical control of previous Finnish Myeloma Group study with lenalidomide, bortezomib, and dexamethasone induction, followed by ASCT and lenalidomide maintenance [22].

Statistical analyses were carried out using appropriate computer software (IBM SPSS Statistics 26, Chicago, IL, USA). Descriptive statistics are presented in frequencies and percentages for categorical variables. Continuous numerical variables are described using medians with ranges. Comparisons of continuous variables between the two cohorts were tested by Mann–Whitney U-test or *t*-test and comparisons of nominal data were performed using the Pearson’s chi-square or Fisher’s exact tests. Kaplan–Meier’s methodology and the log-rank test were used for survival analyses, for which the day of the inclusion was kept as the starting timepoint. The competitive risk model was used to study associations of detailed risk groups with PFS. In this model, competitive events for withdrawals of the study were death, excess toxicity, or patient’s/physician’s decision. Multivariate analysis for PFS was conducted using the Cox regression model. A two-tailed *p* value of <0.05 was considered significant.

### 2.5. Ethics

The study was approved by the Finnish Medicines Agency, the Norwegian Medicines Agency, the Swedish Medical Products Agency and the State Medicines Control Agency of Lithuania and is registered at ClinicalTrials.gov (identifier NCT03376672) on 13 December 2017. The study was conducted in accordance with the Declaration of Helsinki. Ethics committee registration numbers are 199/2016 in Finland, P-17-52 in Lithuania, 2016/1361 in Norway and 2017/2392-31/1 in Sweden. Written informed consent was obtained from all participating patients.

## 3. Results

Between May 2018 and March 2020, altogether 120 NDMM patients were enrolled in the study. Based on FISH aberrations, 57 patients (47.5%) were assigned to the HR group and 63 patients (52.5%) the NHR group. Altogether, 41 patients had 1q gain, and in 30 of them, 1q gain was the only HR cytogenetic finding. Twenty patients had three copies of 1q gain, five had four copies, and the number of copies in sixteen patients is unknown. The characteristics and demographics of the patients are presented in Table 1.

Altogether, 111 patients (93%) completed induction, 101 (84%) proceeded to ASCT, 96 (80%) received consolidation, and 91 (76%) proceeded to maintenance (Figure 1).

There was no significant difference between the HR and NHR groups regarding dose intensity, engraftment time or hospitalization. The mobilization and collection data are presented in Appendix A.

Ninety-one of the transplanted patients (90%) proceeded to maintenance therapy. Regarding the dose intensity during maintenance, ixazomib at a dose of 4 mg for three days per cycle was given in 77% of cycles, ixazomib 3 mg for three days per cycle in 5% of cycles and ixazomib 2.3 mg in 5% of cycles. The remaining cycles were conducted with fewer days of ixazomib. In the HR group, 11% of the cycles (three patients) were conducted without ixazomib due to neuropathy. During maintenance, lenalidomide dosing in the HR group was 10 mg in 29% of the cycles and 15 mg in 52% of the cycles, and the dose was decreased 5 mg in 7% of the cycles due to toxicity. In the NHR group, the patients received lenalidomide at a dose of 10 mg in 23% of the cycles, 15 mg in 64% of the cycles and 5 mg in 13% of the cycles due to toxicity.

After induction, the ORR was 87%. At least CR was detected in 8 patients (7%) after induction, in 24 patients (20%) after ASCT and in 32 patients (27%) after consolidation. At one year on maintenance, 46 patients (38%) and at two years on maintenance 43 patients (36%) had CR or better.

At two years on maintenance, 66 patients (55%) were still on the study. The treatment responses during the study course are presented in Table 2.

### 3.1. Flow-MRD Status

The median number of TNCs analyzed by flow-cytometry was 4.7 × 10^6^ (range 0.9−14.9 × 10^6^). There was no difference in the analyzed amount of TNCs between the risk groups. In the whole population, 307 out of 344 (89%) of the analyzed samples fulfilled the definition of good quality (>2 × 10^6^ cells in a sample). Also, the median lower limit of detection was comparable between the groups 4 × 10^−6^ (1 × 10^−6^–1.9 × 10^−5^) vs. 4 × 10^−6^ (1.5 × 10^−6^–4 × 10^−5^), *p* = 0.180.

At any time during protocol treatment, 28% (34/120) had MRD < 10^−5^ at least once. Regarding HR patients with CR or better, 33% (3/9) of double-hit patients and 50% (15/30) of those with one adverse cytogenetic factor had undetectable MRD < 10^−5^ at least once. At 2 years on maintenance, 36% (43/120) had CR or better and 42% (18/43) had MRD < 10^−5^. Sustained undetectable MRD (at least two consecutive samples of MRD < 10^−5^ analyzed every 12 months) was detected in 23% (13) of patients in the HR group compared to 16% (10) of patients in the NHR group, *p* = 0.573. Loss of undetectable MRD was defined as a positive MRD result after one MRD < 10^−5^.

### 3.2. Progression-Free Survival

Altogether, 23% (28) of patients progressed until two years on maintenance. In total, 45% (54) of patients dropped out of the study before the timepoint of analysis. According to the competitive risk model, PFS at 2 years from induction did not differ between the groups: 0.82 (95% CI 0.71–0.92) in the NHR group vs. 0.74 (95% CI 0.57–0.91) in the 1q gain-only HR risk group vs. 0.56 (95% CI 0.27–0.86) in the HR with 1q gain group vs. 0.87 (95% CI 0.69–1.04) in the HR without 1q gain group, *p* = 0.696. Of note, when PFS was compared between the HR group without 1q gain (*n* = 27; 23%) and the NHR group (*n* = 93; 77%), no significant differences were found either (*p* = 0.579). There was no statistical difference when the influence of the high-risk cytogenetic abnormalities (t(4;14), t(14;16), t (14;20), 1q+ or del (17 p) on PFS was analyzed (*p* = 0.859).

According to Kaplan–Meier analysis, PFS at two years on maintenance did not differ between the groups (Figure 2). At two years on maintenance PFS rates were 66% in the HR group vs. 76% in the NHR group. Of note, 95% of the patients with sustained MRD < 10^−5^, 82% of the patients who turned MRD-positive and 61% of MRD-positive patients had no disease progression at two years on maintenance (*p* < 0.001) (Figure 3). On Cox regression analysis (univariate), R-ISS score and undetectable MRD <10^−5^ were significant prognostic factors for PFS (Appendix A). PFS rates for different HR subgroups are presented in Appendix A.

### 3.3. Overall Survival

The median follow-up time was 34 (range 1–53) months. At cutoff, 109 patients were alive: 50 (88%) in the HR group vs. 59 (94%) in the NHR group. Treatment-related mortality during the study course up to two years on maintenance was 1.7%. One patient died due to pulmonary embolism and another patient died for unknown reasons. Eight patients have died after leaving the study and one patient after two years on maintenance. OS rate after two years on maintenance was 89% in the HR group vs. 92% in the NHR group (*p* = 0.624), respectively. Patients with sustained undetectable MRD < 10^−5^ also had comparable OS between risk groups (*p* = 0.252) (Appendix A), and median OS was not reached in either group.

### 3.4. Adverse Events

Grade 3–4 hematologic toxicity, especially neutropenia, was the most common reported adverse event and was especially prevalent during maintenance (30/120 patients; 30%), albeit a great majority of those patients were asymptomatic and had no need for hospitalization. Any-grade thrombocytopenia was detected in 10 patients during maintenance, and 40% of those had at least grade 3 toxicity. Maculopapular rash appeared principally before maintenance in 39/120 (32%) of the patients, but only 9/120 patients (8%) had severe skin reactions of grade 3 or worse. Elevated liver enzymes, mainly low-grade, were seen in 27/120 patients (22%), whereas grade 3 toxicity or worse was noted in 6/120 (5%) of the patients. Moderate sensory neuropathy was detected more often than motor neuropathy (at any grade, 31/120 (25%) in HR vs. 8/120 (6%)) in the NHR group. The most common grade 3 or worse infection was pneumonia, especially during maintenance (4 patients in the HR group vs. 6 patients in the NHR group). Altogether, seven patients were withdrawn from the study due to toxicity or SAEs (multiorgan failure, encephalitis, putamina infarct, diarrhea, skin lesion, toxicity due to mobilization chemotherapy, peripheral neuropathy). The specified data regarding numbers of patients with AEs reported are presented in Table 3.

Regarding secondary primary malignancies (SPMs), two patients (1.7%) were reported, one with pancreas cancer with liver metastases and one with a heavy smoking history who died of lung cancer four days after being withdrawn from the study.

## 4. Discussion

This prospective multicenter study evaluated the flow MRD < 10^−5^ after IRD induction plus ASCT followed by IRD consolidation and risk-based maintenance with lenalidomide ± ixazomib. The study also aimed to analyze the impact of combining lenalidomide with ixazomib maintenance on PFS and OS in high-risk patients. Flow MRD < 10^−5^ was achieved in 28% (34/120) of the patients during the study course. We also found that irrespective of MM risk stratification, altogether 95% of the patients with sustained MRD < 10^−5^, 82% of the patients who turned MRD-positive and 61% of those who never reached undetectable MRD had no disease progression at two years on maintenance. Most importantly, HR cytogenetics had no influence on sustained undetectable MRD or PFS.

In the era of novel agents, the best induction and maintenance regimen or the ideal length of maintenance after ASCT is still under debate. The CASSIOPEIA study reported over 90% ORR after four-drug induction of daratumumab, bortezomib, thalidomide and dexamethasone (D-VTD) [23]. Also, the most recent IFM KRd study with prolonged carfilzomib–lenalidomide–dexamethasone exposure with one-year lenalidomide maintenance produced an even higher ORR of 97%, and 66% of the patients had undetectable MRD < 10^−6^ [24]. However, a large registry study from the Center for International Blood and Marrow Transplant Research (CIBMTR) database concluded VCD induction to be noninferior, with VRD induction on multivariate analysis regarding treatment response and patient survival [25] in concordance with the results of the randomized EVOLUTION study [26]. Contradictory statements have also been presented concerning the benefit of lenalidomide-based induction regarding the better ORR and even OS [27,28]. A pooled analysis of quadruplets including the CASSIOPEIA and GRIFFIN studies concluded that adding CD38 antibody to induction treatment produced deeper responses compared with triplets [29], albeit longer follow-up data of those first-line studies are awaited.

Ixazomib, an oral proteosome inhibitor, is approved after at least two previous treatment lines in an all-oral triplet with lenalidomide and dexamethasone. A phase 2 IFM study (2013-06) with IRD induction and fixed-length ixazomib maintenance produced over 90% 3-year OS in transplant-eligible patients [3]. However, they found PFS in that combination to be inferior to VRD + ASCT + lenalidomide maintenance [3]. The result of the phase 2 HOVON 143 trial for non-eligible NDMM patients consisting of ixazomib, daratumumab and low-dose dexamethasone was not encouraging, with a median PFS of 18.2 months [30]. In addition, a phase 1/2 study with twice-weekly administration of ixazomib with RD was associated with excess toxicity and proved not to be effective [31]. In turn, in the trial by Rosinol et al., IRD vs. RD maintenance after VRD + ASCT + VRD consolidation program led to comparable PFS [32]. Based on those findings and as no randomized trials comparing IRD induction to other two- to three-drug combinations exist, there are insufficient data for IRD to be raised as induction treatment before ASCT. However, in the present trial, the 3-year OS was 86% in the HR group compared to 83% of all patients in a previous Finnish trial [22], suggesting that double maintenance with ixazomib and lenalidomide could benefit HR patients. This finding highlights the importance of risk-based proteasome inhibitor use in maintenance. Also, according to the results of the CIBMTR study, post-transplant maintenance seemed to be more important than the induction regimen used for outcome, especially in those having at least CR after ASCT [33]. Nowadays, lenalidomide as maintenance is reimbursed based on the evidence of both PFS [34] and even OS benefit [18]. The guidelines of the Mayo Clinic [35] recommend prolonged treatment with bortezomib and lenalidomide in HR patients due to early progress compared to NHR patients. It is worth considering treating HR patients with prolonged PIs plus IMiDs, balancing tolerability and benefits, taking into account that there are some reports on increased mortality during ixazomib maintenance [36]. Of note, if the all-oral IR combination were to be reimbursed, the compliance might improve regarding the use of double maintenance with IR compared to the use of parenteral medication, e.g., in the case of long distances to treatment centers.

A previous single-arm study suggested the need for dose modifications in a proportion of patients during maintenance with ixazomib caused principally by hematologic toxicity [37], in concordance with the findings of the more recent IFM 2013-06 study [3]. In the randomized double-blind TOURMALINE-MM3 study, the patients with ixazomib exposure had comparable safety profile to the placebo group, and the most common grade 3 side effects reported were in addition to infections (14%)—also hematologic toxicity, especially thrombocytopenia (5%), diarrhea (3%) and rash (2%)—resulting in ixazomib dose reductions [5]. We found grade 3 or worse neutropenia in the 30% of patients during maintenance, which was more likely caused by lenalidomide based on previous data of ixazomib side effects in the TOURMALINE 1 study [38]. However, toxicity of grade 3 or worse—thrombocytopenia (11%), neuropathy (6%) and gastrointestinal disorders (5%)—due principally to the ixazomib use was detected in the present analysis, resulting in ixazomib dose reductions. Discontinuation of the treatment due to excess toxicity was seen in 6% of patients in the present study, in line with a previous study [22].

Rawstron et al. concluded in a randomized phase 3 study that a log depletion in MRD by flow cytometry assay was associated with a one-year survival benefit [39]. However, the most practical and sensitive method to assess MRD before and after ASCT in patients with MM is still under debate. Most published data have reported associations with prolonged PFS and undetectable MRD up to a detection limit of 10^−5^ [7]. In the PETHEMA/GEM2014MAIN trial, after longitudinal measuring of MRD, 82% risk reduction of disease progression in patients with undetectable MRD was suggested, regardless of the timepoint of analysis [40]. However, MRD kinetics are an even more important prognostic marker than a single timepoint result of undetectable MRD [7].

Sustained undetectable MRD as a relevant endpoint during maintenance has been presented [7]. In the present study, sustained undetectable MRD < 10^−5^ significantly prolonged PFS despite cytogenetic risk, in line with previous findings [7]. Compared to the previous FMG MM02 study [22], the proportion of undetectable MRD cases was not remarkably better (28% vs. 21%), possibly due to limited efficacy of the IRD combination in induction and consolidation. In the future, in addition to patient- and disease-specific factors, a sustained undetectable MRD could possibly result in therapy decisions regarding in particular the question of fixed-time vs. continuous maintenance and the role of double maintenance in HR patients.

There are some limitations in this study. First, a proportion of the patients received consolidation cycles in the present study before ASCT due to the COVID-19 epidemic, which may have an influence on analyses. However, in some centers, six induction cycles before ASCT might be normal practice. Second, we are lacking information of copy numbers of 1q gain in 34% of the patients. The strengths of the study are the prospective setting, longitudinal data, and several consecutive samples for MRD analysis. In addition, the potency of the study is the sensitive MRD analysis, with a median of >4 million TNCs analyzed at the consecutive timepoints including two years on maintenance with novel drugs.

## 5. Conclusions

To conclude, all-oral ixazomib-based induction, consolidation, and risk-based maintenance with lenalidomide ± ixazomib was completed with reasonable adverse effects. At 2 years on maintenance, 36% of the patients had CR or better and 42% of those had undetectable MRD < 10^−5^. HR cytogenetics had no influence on sustained undetectable MRD or PFS. Prolonged all-oral ixazomib plus lenalidomide maintenance might improve PFS in HR patients.

## Figures and Tables

**Figure 1 cancers-16-01024-f001:**
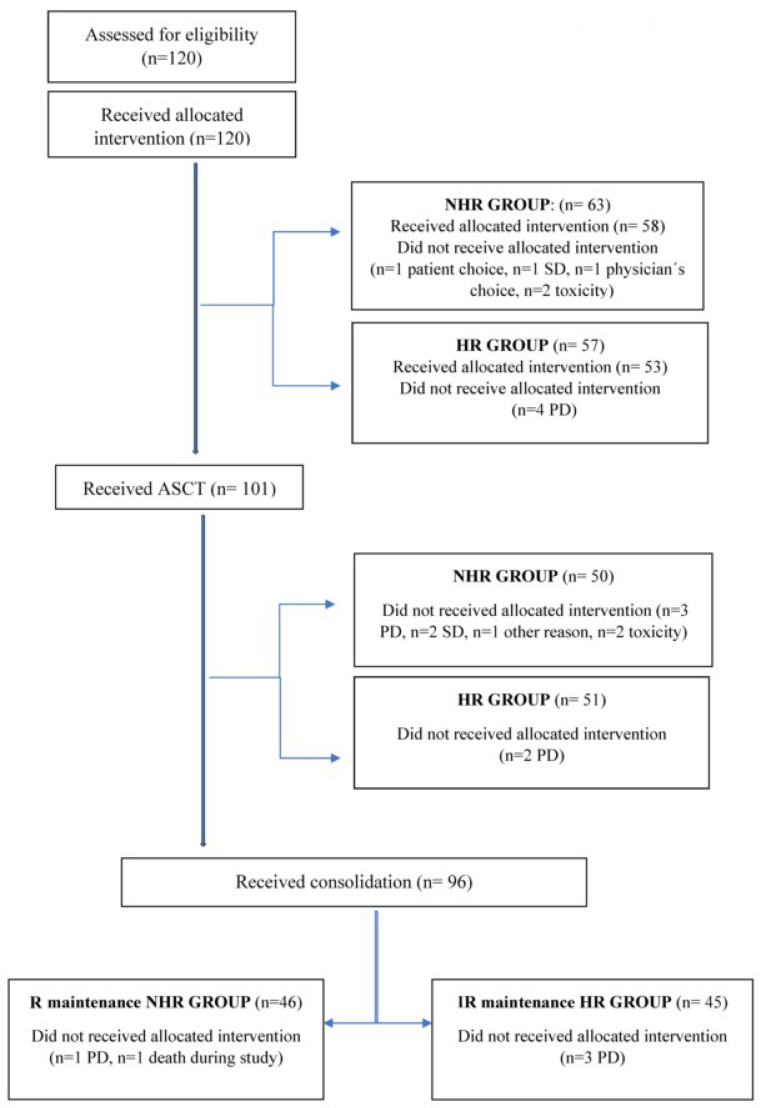
Flow diagram of the study course of 120 MM patients.

**Figure 2 cancers-16-01024-f002:**
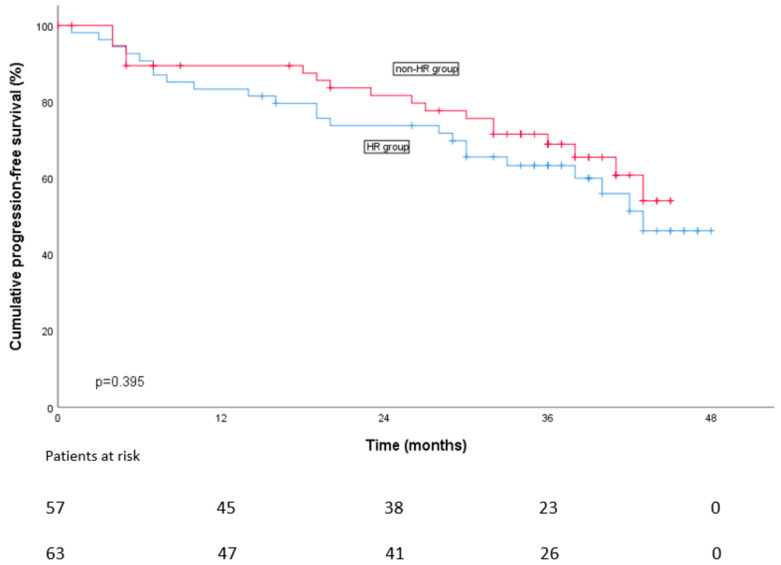
Progression-free survival according to two risk groups (ITT).

**Figure 3 cancers-16-01024-f003:**
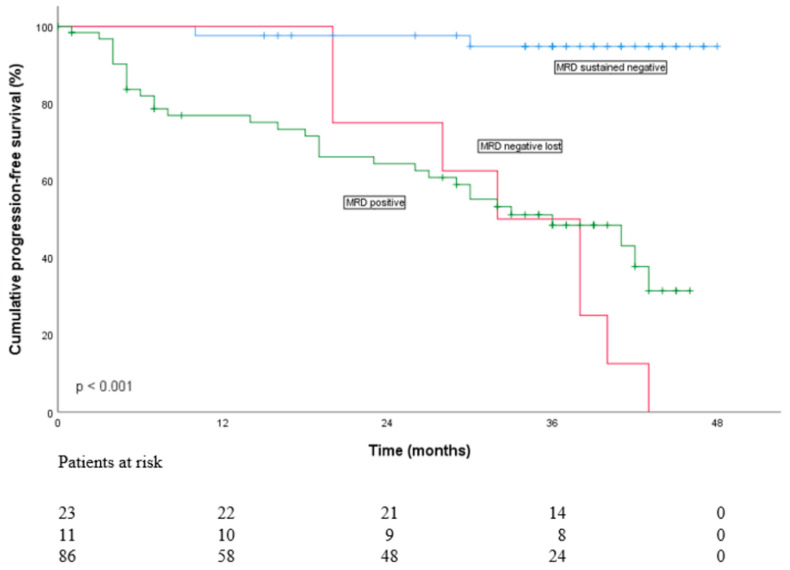
Impact of MRD status < 10^−5^ on progression-free survival (ITT).

**Table 1 cancers-16-01024-t001:** Characteristics and demographics of 120 patients with myeloma according to risk groups.

Variable	HR Group n = 57 (%)	NHR Group n = 63 (%)	Significance *p*
Age	63 (41–70)	59 (40–70)	0.089
Gender			0.228
Female	27 (47)	23 (37)	
Male	30 (53)	40 (63)	
ECOG			0.103
0	33 (58)	34 (54)	
1	17 (30)	27 (43)	
2	7 (12)	2 (3)	
Heavy chain type †			0.432
IgG	32 (56)	43 (68)	
IgA	14 (25)	12 (19)	
IgD	1(2)		
Light chain type			**0.019**
Kappa	30 (53)	47 (75)	
Lambda	27 (47)	16 (25)	
IMWG risk group *			**<0.001**
low	1 (2)	18 (27)	
standard	42 (74)	39 (62)	
high	14 (24)	2 (3)	
R-ISS			**<0.001**
1	9 (16)	31 (49)	
2	42 (74)	26 (42)	
3	6 (10)	2 (3)	
FISH findings ‡			
del 13q/-13	16 (28)	7 (11)	
del 17p	10 (18)	3 (5)	
+1q	42 (74)		
t(4;14)	14 (25)		
t(11;14)	10 (18)	11 (18)	
t(14;16)	4 (7)		
t(6;14)	1 (2)		
t(14;20)	3 (5)		
Treatment indication			0.055
CRAB criteria positive	51 (89)	62 (98)	
Biomarker-based	6 (11)	1 (2)	

Abbreviations: † 10 patients in HR group and 8 patients in NHR group had no heavy chain type; * IMWG was not analyzed in 4 patients in NHR group; ‡ all the patients with t(11;14) in the HR group also had high-risk cytogenetic finding and eight of those had +1q, one patient had +1q plus del17p (cut off value 60%) and one patient had t(14,16); Bolded number means statistically significant (*p* < 0.05).

**Table 2 cancers-16-01024-t002:** Responses after induction, post-ASCT, after consolidation and 1 and 2 years on maintenance (ITT) according to the risk group.

	Post-Induction	Post-ASCT	After Consolidation	1 Year on Maintenance	2 Years on Maintenance
	n = 120 (%)	n = 120 (%)	n = 120 (%)	n= 120 (%)	n = 120 (%)
sCR	2(2)	6 (5)	11 (9)	18 (15)	22 (18)
CR	6 (5)	18 (15)	21 (18)	28 (23)	20 (17)
VGPR	34 (28)	34 (28)	38 (32)	24 (20)	17 (14)
PR	59 (49)	38 (32)	21 (18)	11 (9)	7 (6)
SD	7 (6)				
PD	9 (8)	3 (3)	4 (3)	7 (6)	12 (10)
Out other cause †	3 (3)	2 (2)		3 (3)	1 (1)
Cumulative out		19 (16)	24 (20)	29 (24)	40 (33)
Death			1 (1)		1 (1)

Abbreviations: ASCT = autologous stem cell transplantation; CR = complete remission; ITT = intent-to-treat; MRD = measurable residual disease; PD = progressive disease; PR = partial response; sCR = stringent complete remission; SD = stabile disease**;** † Includes six cases due to toxicity, one case due to physician´s choice, one case due to patient´s choice and one protocol violation.

**Table 3 cancers-16-01024-t003:** Number of patients with reported SAEs during the study course.

AE	Both Groups n = 120		HR Group n = 57			NHR Group n = 63		
	Grade ≥ 3 during Ind n (%)	Grade ≥ 3 during Cons n (%)	Grade 1 during Maint n (%)	Grade2 during Maint n (%)	Grade 3 during Maint n (%)	Grade 1 during Maint n (%)	Grade 2 during Maint n (%)	Grade 3 during Maint n (%)
HEMATOLOGIC								
Anemia	2 (2)		3 (5)		1 (2)	2 (3)		1 (2)
Neutropenia	10 (8)	9 (8)		1 (2)	15 (26)			20 (32)
Thrombopenia		2 (1)	1 (2)	4 (7)	4 (7)		1 (2)	
NON-HEMATOLOGIC								
Fever	10 (8)	2 (2)			2 (4)			2 (3)
Pneumonia	5 (4)			4 (7)				6 (10)
Septicemia	3 (3)							
Upper respiratory tract infection	3 (3)		3 (5)	5 (9)	1 (2)	4 (6)	4 (6)	1 (2)
Covid19			1 (2)		1 (2)	1 (2)	1 (2)	2 (3)
Gastrointestinal disorders	1 (1)	1 (1)	3 (5)	3 (5)	1 (2)	1 (2)	5 (8)	
Nausea/fatigue	2 (2)		5 (9)	1 (2)			1 (2)	1 (2)
Elevated ALT	3 (3)		3 (5)	2 (4)	2 (4)	4 (6)		1 (2)
Acute renal failure	3 (3)	1 (1)						
Skin reactions	8 (7)	1 (1)	2 (4)	2 (4)			2 (3)	
Deep venous thrombosis						1 (2)	3 (5)	
Pulmonary embolism	1 (1)	1 (1)						1 (2)
Arterial thrombosis eye					1 (2)			
Peripheral neuropathy sensory	2 (2)	1 (1)	4 (7)	3 (5)	1 (2)	1 (2)	1 (2)	2 (3)
Peripheral neuropathy motor	1 (1)		2 (4)	2 (4)				1 (2)
Cardiac/arrythmia	1 (1)							1 (2)
Muscle pain		1 (1)	1 (2)	2 (4)	1 (2)	1 (2)		1 (2)
SPMs					2 (4)			

## Data Availability

The datasets created and analyzed during the study are available from the corresponding author on reasonable request.

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
