# Peer review of "Ixazomib, Lenalidomide, and Dexamethasone (IRD) Treatment with Cytogenetic Risk-Based Maintenance in Transplant-Eligible Myeloma: A Phase 2 Multicenter Study by the Nordic Myeloma Study Group"

_cancers, 2024, doi:10.3390/cancers16051024_

Round 1

Reviewer 1 Report

Comments and Suggestions for Authors

The authors report a frontline study in 120 patients with NDMM, which uses IRD backbone as therapeutic strategy. This is a similar design to the Tourmaline MM-2, which though focuses on transplant ineligible patients.

- What was the reason for no transplant or maintenance in a proportion of patients? Were these patients included in the PFS/OS analysis?

- Even though the numbers might be small, any specific high-risk CA benefitting more or less from this approach?

- Can the authors include a forest plot as a figure with the variables?

- How do the authors envisage to incorporate this regimen in the clinics? The opportunity of an all oral regimen could be especially appropriate in certain patient groups or geographic locations (e.g. less travels to the hospital, etc). Please include this aspect in the discussion. 

Comments on the Quality of English Language

Some typos and errors in punctuation.

Reviewer 2 Report

Comments and Suggestions for Authors

The Authors submit a manuscript entitled “ Ixazomib, Lenalidomide and Dexamethasone (IRD) Treatment 2 with Cytogenetic Risk-Based Maintenance in 3 Transplant-Eligible Myeloma: A Phase 2 Multicenter Study by 4 the Nordic Myeloma Study Group”.

The primary objective of the study is reported to be the role  of double maintenance with ixazomib and lenalidomide in cytogenetic high risk (HR) patients.

However , the primary objective of the study is not in line with the primary endpoint, reported as the minimal residual disease (MRD5), at any time. 

  The Authors should describe the results of treatment in terms of response and MRD till the beginning of maintenance in all patients and in the NHR and HR groups separately, and then analyze the cumulative relapse incidence and loss of MRD5 in the HR patients treated with I+L and in NHR treated only with L.

Comments on the Quality of English Language

English has to be improved

Reviewer 3 Report

Comments and Suggestions for Authors

The authors reported IRD treatment with cytogenetic risk-based maintenance in transplant-eligible myeloma patients. These data seem to be precious. I have some comments.

1.  Cytogenetic risk-based maintenance therapy is thought to be a novel strategy and strength point for this manuscript. However, the meaning of " Cytogenetic risk-based maintenance" is difficult to understand at a glance. Therefore, I suggest below.

In abstract, the description "Lenalidomide±ixazomib" should be replaced with other description so that many readers can understand the maintenance therapy that IXA+R for high risk patients,and R for non-high risk patients.

In Figure1, Lenalidomide, and Lenalidomide+ixazomib should be added to the maintenance boxes at the bottom.

2. Table2 should be separately listed by high risk and non-high risk group.

3. Figure2,3 and supple Fig1,2

   Number at risk should be add at the bottom of figure.

4. Table1, EGOC → ECOG

Reviewer 4 Report

Comments and Suggestions for Authors

The original article “Ixazomib, Lenalidomide and Dexamethasone (IRD) Treatment with Cytogenetic Risk-Based Maintenance in Transplant-Eligible Myeloma: A Phase 2 Multicenter Study by the Nordic Myeloma Study Group” reported that the survival time between the HRCA cases treated with IR and non-HRCA cases treated R maintenance therapy was similar, and sustained MRD negativity predicted long PFS but loss of MRD-negativity led to poor PFS similarly to those with MRD-positivity. This prospective study is very interesting for readers even after anti-CD38 monoclonal antibody is used as induction and post-ASCT treatment. However, there were several minor issues.

1.     For screening of cytogenetic abnormality, was CD138 purification done? If so, the cutoff value for HRCA excluded del17p might be low.

2.     After CR was achieved, was imaging study, such as low-dose body CT, PET/CT, or whole-body MRI repeated to detect new bone diseases and plasmacytomas similarly to MRD using flowcytometry in bone marrow sample?   

3.     Was the treatment response after induction therapy affect the clinical outcome of maintenance therapy, especially the HRCA cases treated with IRd?

4.     10 patients with t(11;14) were categorized as HRCA group. Did these patients have del17p or 1q21 abnormality?

5.   The number of HRCA patients with CRAB symptom seemed to be lower than those in the non-HRCA patients with CRAB symptom. Do the author consider that these difference of patient characteristics affect clinical outcomes? 
